# Impact of 2018 ESC/ESH and 2017 ACC/AHA Hypertension Guidelines: Difference in Prevalence of White-Coat and Masked Hypertension

**DOI:** 10.3390/healthcare8020122

**Published:** 2020-05-03

**Authors:** Byong-Kyu Kim, Moo-Yong Rhee

**Affiliations:** 1Division of Cardiology, Department of Internal Medicine, Dongguk University College of Medicine, Gyeongju Hospital, Gyeongju 38067, Korea; bleumatin@dongguk.ac.kr; 2Cardiovascular Center, Dongguk University Ilsan Hospital, 27 Dongguk-ro, Ilsandong-gu, Goyang-si, Gyeonggi 10326, Korea

**Keywords:** hypertension, guidelines, white coat hypertension, masked hypertension, ambulatory blood pressure

## Abstract

Our study evaluated whether there were differences in the prevalence of white-coat hypertension (WH) and masked hypertension (MH) based on the 2018 ESC/ESH and 2017 ACC/AHA hypertension guidelines in Korea. The motivation was the lowering of the diagnostic threshold for hypertension in the 2017 ACC/AHA guidelines. Of 319 participants without antihypertensive drug history and with suspected hypertension based on outpatient clinic blood pressure (BP) measured by physicians, 263 participants (51.6 ± 9.6 years; 125 men) who had valid research-grade office BP and 24-h ambulatory BP measurements were enrolled. WH prevalence based on daytime ambulatory BP among normotensive individuals was lower with the ESC/ESH guidelines than the ACC/AHA guidelines (29.0% vs. 71.4%, *p* < 0.001). However, MH prevalence based on daytime ambulatory BP among hypertensive individuals was higher based on the ESC/ESH guidelines (21.6% vs. 1.8%, *p* < 0.001). Seventy percent of WH cases (2017 ACC/AHA guidelines) and 95.2% of MH cases (2018 ESC/ESH guidelines) occurred in individuals with systolic BP of 130–139 mmHg and/or diastolic BP of 80–89 mmHg. The diagnostic threshold of the 2017 ACC/AHA guidelines yielded a higher prevalence of WH compared to that of the 2018 ESC/ESH guidelines. However, the prevalence of MH was higher with the 2018 ESC/ESH guidelines than with the 2017 ACC/AHA guidelines. The high prevalence of WH and MH in people with a systolic BP of 130–139 mmHg or diastolic BP of 80–89 mmHg suggests the need for a more active out-of-office BP measurement in this patient group.

## 1. Introduction

Traditionally, office systolic blood pressure (BP) 140 mmHg and diastolic BP 90 mmHg have been used as diagnostic thresholds of hypertension. However, the American College of Cardiology/American Heart Association released new hypertension guidelines in 2017 (2017 ACC/AHA) and lowered the diagnostic threshold of office BP to 130/80 mmHg [1]. ACC/AHA also set a new diagnostic threshold for ambulatory BP (125/75 mmHg for 24 h ambulatory BP and 130/80 mmHg for daytime ambulatory BP). However, the 2018 hypertension guidelines of the European Society of Cardiology/European Society of Hypertension (2018 ESC/ESH) retained the previous diagnostic criteria for office BP (office BP: 140/90 mmHg) and ambulatory BP (130/80 mmHg for 24 h ambulatory BP and 135/85 for daytime ambulatory BP) for the diagnosis of hypertension [2]. The major issue raised by the 2017 ACC/AHA hypertension guidelines is an increase in the prevalence of hypertension compared to previous guidelines and the 2018 ESC/ESH guidelines [3,4,5]. However, the impact of different diagnostic thresholds of these guidelines on the prevalence of hypertension phenotypes has not been established.

Recent guidelines have emphasized the use of out-of-office BP measurement to diagnose white-coat hypertension (WH) and masked hypertension (MH) [1,2]. WH is associated with relatively less risk for cardiovascular events or all-cause mortality compared to sustained hypertension [6,7,8,9]. Therefore, antihypertensive drug treatment may be considered for WH patients with high cardiovascular risk, but not for all WH patients [2]. However, in practice, people with WH are more likely to be prescribed BP-lowering drugs because they are misinterpreted as having sustained hypertension. On the contrary, MH has been associated with a similar cardiovascular risk compared to sustained hypertension [6,7,9,10]. Antihypertensive drug treatment should be considered for patients with MH [2]. The prevalence of WH has been reported to be approximately 30–40% in hypertension patients, while the prevalence of MH has been reported as 10–30% [1,2,11]. The prevalence of WH and MH depend on the diagnostic threshold of hypertension [12]. Therefore, changes in the diagnostic threshold of hypertension are likely to change the prevalence of WH and MH. The objective of the present study was to evaluate whether there were differences in the prevalence of WH and MH based on the 2018 ESC/ESH and 2017 ACC/AHA hypertension guidelines.

## 2. Materials and Methods

### 2.1. Study Population

The study population and protocol have been described previously [12,13]. Briefly, individuals whose outpatient clinic BP measured by a physician was ≥140/90 mmHg and who did not have a history of antihypertensive drug treatment were included in the study. The exclusion criteria were as follows: secondary hypertension, hypertensive emergency or urgency, heart failure (New York Heart Association functional class III and VI), clinically significant cardiac arrhythmia, impaired renal function (serum creatinine ≥1.7 mg/dL), pregnancy, night shift work, history of drug or alcohol abuse within 6 months, current participation in other hypertension clinical studies, and prescription of drugs such as steroids, monoamine oxidase inhibitors, oral contraceptives, or sympathomimetic known to affect BP. Altogether, 319 participants who met the eligibility criteria were enrolled and BP measurements according to study protocol were conducted. This study was approved by the Institutional Review Board of the participating hospital. All participants provided written informed consent before enrollment.

### 2.2. Measurement of Office BP and 24 h Ambulatory BP

The BP measurement protocol method has been described previously [12,13]. Briefly, at the first visit, office BP was measured and home BP was measured for 7 days. At the second visit on the eighth day, the office BP was measured and 24 h ambulatory BP measurement was started. The next day, participants visited and completed the 24 h ambulatory BP measurement and then office BP was measured. Research-grade office BP was measured with a validated oscillometric device (WatchBP Home, Microlife, Taipei, Taiwan) three times for each of three visits under the presence of a research nurse after 5 min of seated rest at 1 min intervals. A valid office BP was defined as three measurement readings of three visits (total 9 readings). Ambulatory BP was used as the reference for the diagnosis of hypertension. An automated and non-invasive oscillometric device (Mobil-O-Graph, I.E. M. GmbH, Stolberg, Germany) was used to measure the 24 h ambulatory BP on the non-dominant arm at 30 min intervals. Participants were instructed to continue with their normal daily activities during the study period. A valid 24 h ambulatory BP measurement was defined as valid readings for more than 70% of the total measurement attempts with at least 14 measurements during the daytime (10:00 to 20:00 h) and at least seven measurements during the night time (00:00 to 06:00 h).

### 2.3. Definition of MH and WH

Hypertension and normotension were defined based on the diagnostic threshold of the daytime ambulatory BP of both guidelines. MH was defined as normal BP with office BP measurement and hypertension with daytime ambulatory BP measurement. WH was defined as hypertension with office BP measurement and normal BP with daytime ambulatory BP measurement.

### 2.4. Statistical Analysis

Among the 319 participants, 56 participants were excluded from this study because of invalid office BP or invalid 24 h ambulatory BP data. BP data from 263 participants (138 women and 125 men) were analyzed. The degree of agreement for a diagnosis of hypertension by office BP and ambulatory BP based on the ESC/ESH and ACC/AHA hypertension guidelines was analyzed using Cohen’s kappa value. Sensitivity, specificity, positive predictive value, and negative predictive value for hypertension according to each guideline’s diagnostic threshold of office BP measurement were calculated using daytime ambulatory BP as the reference standard. Further, the differences in the prevalence of MH and WH between the ESC/ESH and ACC/AHA hypertension guidelines were assessed using the Z-test. Statistical analyses were performed using MedCalc for Windows, version 19.0.8 (MedCalc Software; Ostend, Belgium). A *p*-value < 0.05 was considered statistically significant.

## 3. Results

The clinical and demographic characteristics of the studied subjects are described in Table 1.

The prevalence of hypertension by office BP increased to 93.9% (n = 247) based on the diagnostic threshold of 130/80 mmHg (2017 ACC/AHA), from 65.4% (n = 172) based on a diagnostic threshold of 140/90 mmHg (2018 ESC/ESH). The mean differences in systolic and diastolic BP between office BP and daytime ambulatory BP were 3.9 ± 11.0 mmHg and −0.4 ± 8.6 mmHg, respectively.

The ESC/ESH guidelines had lower diagnostic sensitivity and negative predictive value than the ACC/AHA guidelines. However, they showed higher diagnostic specificity and better diagnostic agreement between office BP and ambulatory BP. The kappa value was 0.448 for the ESC/ESH hypertension guidelines and 0.357 for the ACC/AHA hypertension guidelines (Table 2).

Table 3 shows the prevalence of hypertension phenotypes according to the hypertension guidelines. The prevalence of WH among normotensive individuals was lower according to the ESC/ESH guidelines than according to the ACC/AHA guidelines (29.0% vs. 71.4%, *p* < 0.001). However, the ESC/ESH guidelines showed a higher prevalence of MH compared to the ACC/AHA guidelines among hypertensive individuals (21.6% vs. 1.8%, *p* < 0.001).

Majority of participants with MH (95.2%, 40 of 42 individuals) based on the ESC/ESH guidelines and 70.0% of participants with WH (21 of 30 individuals) based on the ACC/AHA guidelines were individuals with office systolic BP of 130–139 mmHg and/or office diastolic BP of 80–89 mmHg (Table 4).

The diagnosis of hypertension based on the diagnostic threshold of 24 h ambulatory BP showed similar results (Appendix A).

## 4. Discussion

In the present study, we found a higher prevalence of WH by applying the 2017 ACC/AHA hypertension guidelines compared to the 2018 ESC/ESH hypertension guidelines. We also noted a higher prevalence of MH according to the 2018 ESC/ESH hypertension guidelines compared to the 2017 ACC/AHA hypertension guidelines.

The 2017 ACC/AHA hypertension guidelines lowered the diagnostic threshold of office BP to 130/80 mmHg from the 140/90 mmHg of previous guidelines. However, the 2018 ESC/ESH hypertension guidelines retained the previous diagnostic threshold of hypertension for office BP. In terms of the diagnosis of WH and MH, out-of-office BP should be measured. Recent guidelines have emphasized the diagnosis of WH and MH using an out-of-office BP measurement such as ambulatory BP or home BP [1,2]. In our study, ambulatory BP was used as a reference diagnosis of sustained hypertension. A consensus diagnostic threshold for ambulatory BP mostly relies on a calculated ambulatory BP value regarding office BP by the distribution of ambulatory BP in a normotensive referenced population [14], regression of ambulatory BP based on office BP [15,16], and analysis of an international database of outcome studies. Therefore, the 2017 ACC/AHA hypertension guidelines lowered the diagnostic threshold of ambulatory BP, considering the lowering of the diagnostic threshold for office BP. However, the 2018 ESC/ESH hypertension guidelines retained the diagnostic threshold of office and ambulatory BP as in previous guidelines. Such a difference in diagnostic thresholds between hypertension guidelines may affect the prevalence of hypertension phenotypes such as MH and WH.

An important aspect of BP control considered in both guidelines is the relevance of out-of-office BP measurement to verify a diagnosis of hypertension and the adequacy of treatment [17]. People with WH have a higher risk of progression to sustained hypertension and cardiovascular events compared to normotensive people [6,10]. Nonetheless, antihypertensive drug treatment is not recommended in WH. This is because no outcome studies are demonstrating the benefits of lowering BP with antihypertensive drugs. However, in real-world practice, WH is problematic as it may lead to the overuse of antihypertensive drugs. Out-of-office BP measurement for the diagnosis of hypertension is not widely used. Occasionally it is difficult to use because of its expensiveness and lack of reimbursement in health care insurance. Therefore, it may be common to misdiagnose WH as sustained hypertension by measuring only office BP, which can lead to excessive treatment in people with low cardiovascular risk. The benefit-harm ratio for BP overtreatment has been shown in recent analyses demonstrating that those with lower baseline cardiovascular disease risk experience more harm than benefit from intensive treatment [18].

Unlike WH, patients with MH are more likely to be under-treated because they are diagnosed as normotensive. However, the risk of cardiovascular events in MH is greater than in normotension or WH and is nears that in sustained hypertension [6,9,10]. As in WH, antihypertensive drug treatment for MH is still questionable. This is because no study has evaluated the effect of antihypertensive drugs on cardiovascular outcomes in MH. In addition, there are no data on the control of out-of-office BP in patients with uncontrolled MH. Recently, a clinical trial was started to find the effectiveness of ambulatory BP-guided treatment for uncontrolled MH [19]. Nevertheless, recent guidelines recommend a stronger consideration of antihypertensive drug treatment to normalize out-of-office BP in patients with MH and uncontrolled MH compared to WH [2].

As shown in the present study, the prevalence of hypertension phenotype varies according to the diagnostic threshold of hypertension. The 2017 ACC/AHA hypertension guidelines may lead to the over-treatment of people with WH while the 2018 ESC/ESH hypertension guidelines may lead to the under-treatment of people with MH. Both guidelines generally agree that subgroups of people who have a 10-year cardiovascular risk >15% will benefit from earlier intervention. Therefore, more active use of out-of-office BP measurements such as ambulatory BP is needed to diagnose WH and MH. The routine measurement of out-of-office BP in all patients may be impractical and cost-ineffective. In the present study, most people with WH according to the 2017 ACC/AHA guidelines (21 of 30 individuals, 70.0%) and a majority of people with MH according to the 2018 ESC/ESH guidelines (40 of 42 individuals, 95.2%) had systolic BP in the range of 130–139 mmHg and/or diastolic BP in the range of 80–89 mmHg. This finding suggests that out-of-office BP measurement should be considered for people within those BP ranges.

The present study has potential limitations. First, we did not analyze the prevalence of WH or MH using home BP as a reference for the diagnosis of hypertension. However, a previous study already reported an increase in WH and a decrease in MH when applying the 2017 ACC/AHA hypertension guidelines of home BP measurement instead of traditional hypertension guidelines [20]. Second, the study population was comprised entirely of Koreans who may have a different lifestyle from other countries, and thus, may not be generalizable to other ethnic groups.

## 5. Conclusions

The diagnostic threshold of the 2017 ACC/AHA guidelines yielded a higher prevalence of WH compared to the threshold of the 2018 ESC/ESH guidelines. However, the prevalence of MH was higher based on the 2018 ESC/ESH guidelines compared to the 2017 ACC/AHA guidelines. A high prevalence of WH and MH in people with systolic BP of 130–139 mmHg or diastolic BP of 80–89 mmHg suggests the need for a more active out-of-office BP measurement in these groups.

## Figures and Tables

**Table 1 healthcare-08-00122-t001:** Characteristics of study participants.

Variables	Values
Number (n)	263
Age, years	51.6 ± 9.6
Sex	
Men, n (%)	125 (47.5)
Women, n (%)	138 (52.5)
Body mass index, kg/m^2^	25.4 ± 3.4
Current smoker, n (%)	51 (19.4)
Alcohol use, n (%)	131 (49.8)
Diabetes, n (%)	29 (11.0)
Office BP	
Systolic BP, mmHg	141.1 ± 12.5
Diastolic BP, mmHg	91.7 ± 9.5
Daytime ambulatory BP	
Systolic BP, mmHg	137.2 ± 14.5
Diastolic BP, mmHg	92.1 ± 12.5
24 h ambulatory BP	
Systolic BP, mmHg	132.8 ± 12.3
Diastolic BP, mmHg	88.6 ± 11.1

Values are shown as mean ± standard deviation, or number and percent in parenthesis, as appropriate. BP = blood pressure.

**Table 2 healthcare-08-00122-t002:** Degree of agreement for the diagnosis of hypertension by office measured blood pressure according to the 2018 ESC/ESH and 2017 ACC/AHA hypertension guidelines. Referenced to daytime ambulatory blood pressure.

	Sensitivity	Specificity	PPV	NPV	Kappa
2018 ESC/ESH guidelines	78.4	71.0	88.4	53.8	0.448
	(71.9–83.9)	(58.8–81.3)	(83.9–91.7)	(46.1–61.3)	
2017 ACC/AHA guidelines	98.2	28.6	87.8	75.0	0.357
	(95.4–99.5)	(15.7–44.6)	(85.6–89.7)	(50.5–89.9)	

PPV = positive predictive value; NPV = negative predictive value; Ref = reference; BP = blood pressure; ESC/ESH = European Society of Cardiology/European Society of Hypertension; ACC/AHA = American College of Cardiology/American Heart Association.

**Table 3 healthcare-08-00122-t003:** Prevalence of white coat hypertension and masked hypertension based on a diagnostic threshold of the 2018 ESC/ESH and 2017 ACC/AHA hypertension guidelines. Referenced to daytime ambulatory blood pressure.

	Normotensives		Hypertensives	
	Normotension n (%)	White-Coat Hypertension n (%)	Masked Hypertension n (%)	Sustained Hypertension n (%)
2018 ESC/ESH criteria	49 (71.0)	20 (29.0)	42 (21.6)	152 (78.4)
2017 ACC/AHA criteria	12 (28.6)	30 (71.4)	4 (1.8)	217 (98.2)
*p* value *	<0.001	<0.001	<0.001	<0.001

Normotensives and hypertensives were diagnosed based on daytime ambulatory blood pressure. * *p*-value from Z-test of the difference between the rate of each phenotype of hypertension according to criteria of the ESC/ESH and ACC/AHA hypertension guidelines. ESC/ESH = European Society of Cardiology/European Society of Hypertension; ACC/AHA = American College of Cardiology/American Heart Association.

**Table 4 healthcare-08-00122-t004:** Prevalence of hypertension phenotypes in participants with a systolic blood pressure of 130–139 mmHg and/or diastolic blood pressure of 80–89 mmHg.

	Normotension n (%)	White-Coat Hypertension n (%)	Masked Hypertension n (%)	Sustained Hypertension n (%)
2018 ESC/ESH criteria	35 (46.7)		40 (53.3)	
2017 ACC/AHA criteria		21 (28.0)		54 (72.0)

ESC/ESH = European Society of Cardiology/European Society of Hypertension; ACC/AHA = American College of Cardiology/American Heart Association.

## Data Availability

The data used to support the findings of this study are available from the corresponding author upon request.

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
