# Peer review of "Impact of 2018 ESC/ESH and 2017 ACC/AHA Hypertension Guidelines: Difference in Prevalence of White-Coat and Masked Hypertension"

_healthcare, 2020, doi:10.3390/healthcare8020122_

Round 1

Reviewer 1 Report

Authors present an interesting study covering answers to some basic questions, especially showing differences in the prevalence of white coat hypertension and masked hypertension. A some Authors previously showed (Poudel B, Booth JN 3rd, Sakhuja S, Moran AE, Schwartz JE, Lloyd-Jones DM, Lewis CE, Shikany JM, Shimbo D, Muntner P. Prevalence of ambulatory blood pressure phenotypes using the 2017 American College of Cardiology/American Heart Association blood pressure guideline thresholds: data from the Coronary Artery Risk Development in Young Adults study. J Hypertens. 2019 Jul;37(7):1401-1410. doi: 10.1097/HJH.0000000000002055) novel ACC/AHA guidelines help to diagnose more white coat hypertension cases, but less masked hypertension. There are not many data analyzing effect of ESC guidelines on this type of hypertension prevalence, so this makes this paper interestring.

However:

1) data may be hard to compare with other studies, why in this study the office blood pressure was measured in 1 minute intervals? Some researchers show results of blood pressure measurement in 30 s intervals (Höller J, Villagomez Fuentes LE, Matthias K, Kreutz R. The Impact of Measurement Methods on Office Blood Pressure and Management of Hypertension in General Practice. High Blood Press Cardiovasc Prev. 2019 Dec;26(6):483-491. doi: 10.1007/s40292-019-00347-z). Also, as mentioned by Höller et al. patients in other studies sometimes do not have 3 repated measurements and do not have a chance to sit minimum 5 minutes in a quiet place.

2) please correct supplemental Table 2: "WH" instead of "HW" on the top of the Table.

Author Response

Response to Reviewer 1 comments

Authors present an interesting study covering answers to some basic questions, especially showing differences in the prevalence of white coat hypertension and masked hypertension. A some Authors previously showed (Poudel B, Booth JN 3rd, Sakhuja S, Moran AE, Schwartz JE, Lloyd-Jones DM, Lewis CE, Shikany JM, Shimbo D, Muntner P. Prevalence of ambulatory blood pressure phenotypes using the 2017 American College of Cardiology/American Heart Association blood pressure guideline thresholds: data from the Coronary Artery Risk Development in Young Adults study. J Hypertens. 2019 Jul;37(7):1401-1410. doi: 10.1097/HJH.0000000000002055) novel ACC/AHA guidelines help to diagnose more white coat hypertension cases, but less masked hypertension. There are not many data analyzing effect of ESC guidelines on this type of hypertension prevalence, so this makes this paper interestring.

However:

  1. data may be hard to compare with other studies, why in this study the office blood pressure was measured in 1 minute intervals? Some researchers show results of blood pressure measurement in 30 s intervals (Höller J, Villagomez Fuentes LE, Matthias K, Kreutz R. The Impact of Measurement Methods on Office Blood Pressure and Management of Hypertension in General Practice. High Blood Press Cardiovasc Prev. 2019 Dec;26(6):483-491. doi: 10.1007/s40292-019-00347-z). Also, as mentioned by Höller et al. patients in other studies sometimes do not have 3 repated measurements and do not have a chance to sit minimum 5 minutes in a quiet place.

Response] There are only a few studies on the measurement interval and number of readings. Landmark trials of hypertension have measured office blood pressure (BP) after 5 min rest, 30 s to 2 min interval, and averaged 2 – 3 readings (Pulse 2018;6:112-123). Based on those landmark trials, most hypertension guidelines recommend taking at least two BP measurements at 1-2 min intervals after several minutes of rest (2003, 2007, 2013, 2018 ESH/ESC guidelines for hypertension, 2017 ACC/AHA hypertension guidelines). Only a few studies have evaluated whether there is a different impact of measurement intervals on the measured BP. Myers et al. (Blood Press Monit 2008;13:333–338) compared BP measurement readings taken at either 1 or 2 min and found no difference. In terms of reading numbers, we found no difference between the average of 2 readings and 3 readings in a previously published paper (Hypertens Res. 2018 Sep;41(9):738-747).

Therefore, data obtained by using different measurement intervals and the number of readings could be compared if they were measured at 30 s to 2 min or more intervals and averaged 2 or more readings. This issue is not the scope of the present research, and we have not added it to the manuscript.

  1. please correct supplemental Table 2: "WH" instead of "HW" on the top of the Table.

Response] Thank you for your careful review. However, we revised the abbreviation according to the suggestion of another reviewer.

Reviewer 2 Report

Thank you for submission of your manuscript. Necessary information might be included in the manuscript but it is not easy to locate.

I found the methods to be slightly confusing. I "think" I understand that there were a total of 319 participants in the study who did have a history of anti-hypertensive drug treatment. Is the reader to assume, from their own calculations, that 56 of the participants had a history of anti-hypertensive drug treatment?

In addition to having their BP taken in the office, did they wear the special BP measurement device throughout the day (24 hour period): is that correct? There is not a clear delineation of the numbers of study participants in the methods section.

In the abstract and the results section it was indicated that there were a total of 319 participants and 125 of them were men. I did not know the ethnicity of the participants until the last sentence of the manuscript. There should have been some mention that the participants were Korean or mention of the country (Korea) in the title and/or the abstract. Additionally, the reader would have to calculate the number of women who were study participants: the reader should not have to do this calculation. 

Results - It was very difficult to follow and determine the differences between those with and without anti-hypertensive drug therapy (I did not see this differentiation: if it is presented it is difficult to locate the information). Also, what is the purpose of singling out the number of men in the study? Are the results just based on data related to the men? Why is there no evidence of differences between men and women?

The tables are not very clear. The captions under the tables are too long therefore, clarity is lost.

Author Response

Response to Reviewer 2 comments

Thank you for submission of your manuscript. Necessary information might be included in the manuscript but it is not easy to locate.

  1. I found the methods to be slightly confusing. I "think" I understand that there were a total of 319 participants in the study who did have a history of anti-hypertensive drug treatment. Is the reader to assume, from their own calculations, that 56 of the participants had a history of anti-hypertensive drug treatment?

Response] We apologize for the confusion. Altogether, 319 participants who met inclusion and exclusion criteria were enrolled. As we have described in the study population section, we recruited only individuals who did not have a history of anti-hypertensive drug treatment and those with suspected hypertension because their blood pressure was ≥ 140/90 mmHg according to a physician’s measurement at the outpatient clinic. Among 319 participants, 56 participants were excluded because of invalid office BP or invalid 24-hour ambulatory BP data. Individuals with a history of anti-hypertensive drugs could not be enrolled in the original study. Therefore, taking anti-hypertensive drugs could not be the reason for exclusion.

To avoid confusion, we described the entire study population, and the study subjects included in this study were described in more detail. Moreover, the definition of valid office BP was added.

  1. In addition to having their BP taken in the office, did they wear the special BP measurement device throughout the day (24 hour period): is that correct? There is not a clear delineation of the numbers of study participants in the methods section.

Response] We apologize for the confusion. As we have described in the previously published papers (Am J Hypertens 2017;30(12):1170-1176, Hypertens Res 2018;41(9):738-747, cited in the manuscript), the original study was designed to compare the diagnosis of hypertension by home BP measurement and ambulatory BP measurement. The BP measurement schedule was described in the previously published paper. In the manuscript submitted previously, we had described the study protocol briefly to avoid redundancy. However, in the revised manuscript, the BP measurement protocol is described in more detail to aid understanding.

To enhance understanding, the figure (see attached file) shows the BP measurement schedule; however, we have not included this figure in the current manuscript to avoid redundancy with our previously published paper (Am J Hypertens 2017;30(12):1170-1176).

  1. In the abstract and the results section it was indicated that there were a total of 319 participants and 125 of them were men. I did not know the ethnicity of the participants until the last sentence of the manuscript. There should have been some mention that the participants were Korean or mention of the country (Korea) in the title and/or the abstract. Additionally, the reader would have to calculate the number of women who were study participants: the reader should not have to do this calculation. 

Response] We apologize for the inconvenience. We have added a phrase of Korean in the abstract. We have added the number of women included in the statistical analysis section and Table 1 in the revised manuscript.

  1. Results - It was very difficult to follow and determine the differences between those with and without anti-hypertensive drug therapy (I did not see this differentiation: if it is presented it is difficult to locate the information). Also, what is the purpose of singling out the number of men in the study? Are the results just based on data related to the men? Why is there no evidence of differences between men and women?

Response] As we have mentioned in a previous response, we did not include participants who have a history of taking anti-hypertensive drugs. This is described as inclusion criteria in the “study population” section. We did not select only men; it was only to indicate the gender distribution of the study population (138 women and 125 men, included in the statistical analysis section and Table 1 of revised manuscript).

  1. The tables are not very clear. The captions under the tables are too long therefore, clarity is lost.

Response] We apologize for the inconvenience. We have included the tables to illustrate our findings more clearly. The tables provided in the current manuscript are the best we could provide after many revisions. In most tables, footnotes are used to indicate the abbreviations used in the table and are difficult to delete. Please consider this difficulty.

Although there are various designs, the table in the manuscript was prepared in a way to best show that the frequency of white-coat hypertension in the normotension population and that of masked hypertension in the hypertension population are statistically different between the diagnostic criteria of both hypertension guidelines when the diagnosis of hypertension was based on the ambulatory BP measurement.
